# The Balance between Differentiation and Terminal Differentiation Maintains Oral Epithelial Homeostasis

**DOI:** 10.3390/cancers13205123

**Published:** 2021-10-13

**Authors:** Yuchen Bai, Jarryd Boath, Gabrielle R. White, Uluvitike G. I. U. Kariyawasam, Camile S. Farah, Charbel Darido

**Affiliations:** 1Peter MacCallum Cancer Centre, 305 Grattan St., Melbourne, VIC 3000, Australia; yuchen.bai@petermac.org (Y.B.); jarryd.boath@petermac.org (J.B.); gabrielle.white@petermac.org (G.R.W.); imalki.kariyawasam@petermac.org (U.G.I.U.K.); camile@oralmedpath.com.au (C.S.F.); 2Australian Centre for Oral Oncology Research & Education, Perth, WA 6009, Australia; 3Sir Peter MacCallum Department of Oncology, The University of Melbourne, Parkville, VIC 3010, Australia

**Keywords:** differentiation, terminal differentiation, oral epithelium, epithelial integrity, epithelial transformation, genetic alterations, oral cancer, therapy response biomarkers

## Abstract

**Simple Summary:**

Oral cancer affecting the oral cavity represents the most common cancer of the head and neck region. Oral cancer develops in a multistep process in which normal cells gradually accumulate genetic and epigenetic modifications to evolve into a malignant disease. Mortality for oral cancer patients is high and morbidity has a significant long-term impact on the health and wellbeing of affected individuals, typically resulting in facial disfigurement and a loss of the ability to speak, chew, taste, and swallow. The limited scope to which current treatments are able to control oral cancer underlines the need for novel therapeutic strategies. This review highlights the molecular differences between oral cell proliferation, differentiation and terminal differentiation, defines terminal differentiation as an important tumour suppressive mechanism and establishes a rationale for clinical investigation of differentiation-paired therapies that may improve outcomes in oral cancer.

**Abstract:**

The oral epithelium is one of the fastest repairing and continuously renewing tissues. Stem cell activation within the basal layer of the oral epithelium fuels the rapid proliferation of multipotent progenitors. Stem cells first undergo asymmetric cell division that requires tightly controlled and orchestrated differentiation networks to maintain the pool of stem cells while producing progenitors fated for differentiation. Rapidly expanding progenitors subsequently commit to advanced differentiation programs towards terminal differentiation, a process that regulates the structural integrity and homeostasis of the oral epithelium. Therefore, the balance between differentiation and terminal differentiation of stem cells and their progeny ensures progenitors commitment to terminal differentiation and prevents epithelial transformation and oral squamous cell carcinoma (OSCC). A recent comprehensive molecular characterization of OSCC revealed that a disruption of terminal differentiation factors is indeed a common OSCC event and is superior to oncogenic activation. Here, we discuss the role of differentiation and terminal differentiation in maintaining oral epithelial homeostasis and define terminal differentiation as a critical tumour suppressive mechanism. We further highlight factors with crucial terminal differentiation functions and detail the underlying consequences of their loss. Switching on terminal differentiation in differentiated progenitors is likely to represent an extremely promising novel avenue that may improve therapeutic interventions against OSCC.

## 1. Introduction

Oral squamous cell carcinoma (OSCC) is the most commonly diagnosed head and neck cancer [1]. OSCC frequently affects the tongue and floor of the mouth within the oral cavity [2] and accounts for approximately 5% of cancer diagnoses in men and 2% in women, worldwide [3]. Among developing countries within Southeast Asia, OSCC represents a quarter of all cancers diagnosed [4], where in India and Sri Lanka, it is the leading cause of cancer death in men [3]. OSCC is the endpoint of a disease spectrum affecting oral epithelial cells and ranging from hyperplasia, dysplasia and carcinoma in-situ, to locally invasive and then metastatic disease. Within the oral cavity, the normal tongue mucosa comprises a connective tissue known as the lamina propria (LP) covered by squamous cell layers forming the oral epithelium (OE). The OE is constantly exposed to acute and chronic environmental insults including damage from mastication, exposure to dietary and airborne antigens, chemical carcinogens, and diverse commensal and pathogenic microorganisms [5]. Tongue OE basal cells attach to the basement membrane and continuously divide to replace damaged cells and to maintain tissue homeostasis. Similar to skin keratinocytes, OE differentiation begins in the basal layer where stem cells differentiate into progenitors, which migrate upward to form the spinous, granular, clear and superficial layers of the tongue, and undergo terminal differentiation to establish a functional protective outer barrier [6,7]. Mitotic figure orientation analyses showed that most daughter progenitor cells divide parallel to the basement membrane and initially remain within the basal layer [8]. Clonal analyses have revealed that up to half of the cells in the basal layer are post-mitotic. Lineage-committed, post-mitotic progenitor cells downregulate the expression of basal layer markers and start to express differentiation-associated genes to exit the basal layer [9]. While they migrate and maturate, differentiated cells upregulate differentiation genes to further differentiate, terminally differentiate and eventually delaminate. Signalling circuitries regulating stem cell fate such as Notch, Hippo and TP63-dependent pathways and DNA and histone methylases are among the main mechanisms that allow stem cells to balance their regenerative potential, while initiating early differentiation programs [10]. The resultant progenitors are cells with potent expansion potential. Factors that end progenitor cell division and induce commitment to terminal differentiation in post-mitotic cells have attracted particular attention as their loss could facilitate the return to a progenitor state through a mechanism activated in response to acute damage or insult [11,12].

The balance between proliferation, differentiation and terminal differentiation is therefore key to maintaining OE homeostasis and securing the physical and physiological functions of the oral barrier [6,13]. Exposure to known risk factors of OSCC such as smoking tobacco and marijuana, betel quid chewing, frequent alcohol use, radiation exposure, immunosuppression, genetic predisposition, poor hygiene and Human Papilloma Virus (HPV) infections may impair the integrity of the OE and facilitate cellular transformation into hyperplastic lesions, dysplasia and even OSCC [1]. Considering its multifactorial origin, it is not surprising that OSCC is a highly heterogeneous disease at the molecular level [14].

## 2. Oral Homeostasis

The OE undergoes a rapid turnover (estimated 4.5 days) to eliminate cells damaged by constant exposure to environmental risk factors. Stem cells initiate this process through their proliferation and differentiation to repopulate the OE and to maintain its homeostasis [15]. Over the last few decades, several techniques have been developed to identify and characterise oral stem cells under different physiological and pathological conditions.

### 2.1. Oral Epithelial Stem Cells

Stem cells are referred to as a quiescent, infrequently dividing cell population with self-renewal properties. The identification and characterization of oral epithelial stem cells have been challenged for decades [16]. The *Bmi1*-expressing OE stem cells were first reported using multi-colour lineage tracing and label retention experiments in the dorsal tongue epithelium [17]. The results suggested that *Bmi1*-positive stem cells were slow-cycling and distributed at a low density within the basal layer. Their proliferation was shown to follow the invariant asymmetry model, supporting an asymmetrical division of long-lived cells that maintain stemness while producing highly proliferative, transit-amplifying daughter cells that differentiate incrementally to produce the clearly defined OE layers. However, using RNAscope on the dorsal tongue and buccal epithelium, Klein’s group found that all basal cells, including progenitors, express *Bmi1*, albeit at low levels, contradicting the data suggesting that *Bmi1* marked rare, solitary stem cells in the suprabasal layers of the OE [9]. Furthermore, through label retention and single-cell sequencing experiments, the authors proposed a population asymmetry model of self-renewal in which half of all basal cells are highly proliferative and half are post-mitotic cells acquiring differentiation programs. In parallel, stochastic proliferation was identified in stem cells within the oral mucosa using the hard palate as a model [18]. Here, the OE is maintained by both symmetrical and asymmetrical divisions of stem cells. Cells expressing *Lrig1* infrequently divide in a symmetrical fashion then become quiescent while *Igfbp5*+ cells undergo rapid asymmetrical division in the lead-up to differentiation. These studies underscore that our understanding of stem cell niche function within the OE is still confounded by various anatomical locations and technical limitations. Spatial single-cell sequencing will be crucial to identify and characterise specific stem cell populations within various head and neck regions and shed light on the mechanisms governing their maintenance and differentiation.

### 2.2. Damage-Induced Stem Cell Activation

The OE is equipped with an effective “catastrophe” response to damage induced by a variety of insults, such as wounding, masticatory stresses, chemical exposure and microbial pathogens. A transcriptional analysis of the OE during wound healing identified reduced differentiation and increased proliferation signatures in OE cells recruited to facilitate tissue repair [19]. This response mechanism is highly effective, and as a result, oral wounds heal rapidly without producing scars. Furthermore, different oral stem cell models show consistent wound healing phases: migration, stem cell expansion and re-epithelization. Upon puncture, irradiation or cytotoxic injuries, OE stem cells were able to migrate to wound-proximal areas, exit quiescence and divide symmetrically to close the wound, simultaneously reducing asymmetrical divisions and restricting differentiation within the OE [18,20]. As soon as proliferating stem cells repopulate the damaged site, a differentiation programme initiates to instate oral homeostasis in injured OE and to re-establish a functional oral epithelial barrier. Cooperative mechanisms linked to the breakdown of the OE basal layer such as the activation of p16^INK4A^ may guide replicative senescence, limiting cellular migration and proliferation while retaining normal growth and differentiation characteristics [21,22,23,24]. In the event of chronic insult, stem cell activation persists in a state that is prone to acquire genetic/epigenetic alterations, which can eventually lead to senescence bypass and oncogenic transformation [25,26,27,28]. Hence, following the damage repair of oral tissue, the differentiation of committed progenitor cells is crucial for tissue remodelling and the prevention of oral diseases and carcinogenesis.

### 2.3. Oral Cancer Stem Cell

Cancer stem cells (CSCs) are believed to act as the drivers of OSCC [29]. These cells are highly tumourigenic with increased self-renewal and clonal capacities. They may originate from the accumulation of multiple oncogenic hits in normal stem cells, in differentiated or senescent cells, which result in their reprogramming into CSC [12,30]. In the context of OSCC, putative CSCs were first described as CD44-expressing cells which formed well-differentiated tumours in a patient-derived xenograft model [31]. CD44 protein expression and spatial regulation has been observed in a dermis-based organotypic 3D culture model of oral epithelium, with an increase in expression from normal OE through to dysplasia and then carcinoma, and with CD44 expression closely associated with the basal layer [32]. In addition to CD44, other characteristics including aldehyde dehydrogenase activation, spheroid-forming ability and the overexpression of CD133 have been proposed as CSC markers in OSCC [33]. Moreover, genetic lineage tracing experiments and carcinogen induced OSCC in mice confirmed the existence of OSCC CSCs with deregulated *Bmi1* expression [34]. Bmi1+ cancer cells can form tumour clones with comparable architectural and molecular features seen in primary tumours. More importantly, slow-cycling Bmi1+ CSCs show increasing invasive and metastatic phenotypes as well as chemoresistance. Additionally, there has been an observed increased trend for the expression of Bmi1 and ABCG2 in dysplastic and malignant tissues compared to normal, highlighting the potential use of these stem cell markers in the malignant transformation of the oral epithelium [35]. Consequently, these CSCs are now recognised as the main drivers of OSCC development and progression, where future therapeutic strategies should aim at targeting their proliferative as well as quiescent states to achieve a complete therapeutic response.

## 3. Molecular Landscape of Oral Epithelial Differentiation

### 3.1. Signalling Pathways

As OE progenitor cells commit to differentiation programs and migrate upward, these cells are exposed to a dynamic gradient of Notch/Wnt ligands and growth factors. These gradients contribute profoundly to the strict regulation of basal-suprabasal compartmentalisation (Figure 1).

### 3.2. Notch Signalling

Notch signalling is regulated predominantly through cell–cell interactions. Notch ligands are expressed in basal cells while Notch receptors are present on suprabasal spinous cells. The activation of Notch signalling in differentiation-committed cells coincides with their detachment from the basement membrane and the expression of terminal differentiation program genes. Notch regulates cell fate commitment and governs the balance between basal progenitor proliferation and supra-basal differentiated cells within a process known as the basal-to-suprabasal switch [36]. In agreement with this, the deregulation of NOTCH signalling in basal cells results in increased tumour susceptibility and a shift towards tumours with poor differentiation [37,38]. This correlates with the proliferation–differentiation imbalance and the *Notch1*-driven terminal differentiation in supra-basal layers [39]. Loss of Notch signalling also synergised with both a *TP53* gain-of-function mutation or the expression of HPV oncogenes to induce high-grade carcinomas within the head and neck region [37]. Moreover, an in vivo CRISPR screen in OSCC identified oncogenic drivers that cooperate with rare mutations in 15 driver genes, all with activities that converge on *Notch* signalling [40]. These results highlight the important role played by Notch signalling in the suppression of OSCC through the promotion of progenitor differentiation. Moreover, aberrant NOTCH signalling was shown to contribute to deregulating the cell cycle, escaping cell death, and establishing a tumour-promoting microenvironment in SCC [41,42]. Recent data from exome sequencing and the Cancer Genome Atlas (TCGA) analyses identified a high frequency for *NOTCH1/2/3* mutations and truncations (~27%) which presumably lead to protein inactivation, disruption of differentiation and OSCC development [42,43,44]. More importantly, mutations of *NOTCH1* are commonly detected in precancerous lesions, highlighting its protective role against the early onset of OSCC in patients [21,40,45]. Contradictory, while *NOTCH1* mutations are mainly considered drivers of the disease, growing evidence points out to the hyperactivation of NOTCH signalling in a subset of OSCC, proposing an oncogenic potential to wild-type *NOTCH1* [40,46]. NOTCH1-targeting γ-secretase inhibitors, which prevent the cleavage of the Notch intracellular domain and consequently block its translocation to the nucleus, are being evaluated in pre-clinical and clinical trials [47,48]. However, no studies to date have evaluated these inhibitors specifically in OSCC, and due to the context and tissue-specific dependent role of NOTCH signalling, clinical application of NOTCH inhibitors may require further caution.

### 3.3. Hippo Pathway

The Hippo pathway controls organ size in *Drosophila* through cell fate determination, tissue regeneration and stem cell self-renewal and is mainly regulated through contact inhibition [49]. In the presence of cell–cell interactions, structural protein complexes activate Hippo kinases on the cell membrane, leading to cytosolic sequestration and proteasomal degradation of the downstream transcriptional factor YAP1 and its co-activator TAZ [50]. This mechanism maintains apical–basolateral polarity for the regulation of stemness and differentiation, and the inactivation of the Hippo pathway perturbs differentiation, promoting proliferation and tissue hyperplasia [49]. In the skin, YAP1 is mainly expressed in basal progenitors and its overexpression expands the proliferative basal cell pool and deregulates terminal differentiation, facilitating tumourigenesis [51]. Indeed, hyperactivation of YAP1 in the mouse oral cavity induces an early onset of OSCC, indicating that this pathway is a potent driver of the disease [52]. Furthermore, 8.6% of OSCC primary tumours show an amplification of the *YAP1* locus and 21.6% with truncated *FAT1* [53], an upstream membrane receptor protein in the Hippo kinase cascade, which results in increased YAP1 activity, malignant progression and poor patient prognosis [54,55]. YAP1 has also been shown to act as a potential biomarker for cetuximab resistance in head and neck cancer [56].

### 3.4. TP63-Regulated Transcription

TP63 belongs to the TP53 family of transcription factors and has two distinct isoforms: TAp63 and ΔNp63. TAp63 possesses a transactivation domain at N-terminal, while ΔNp63 contains a shorter activation domain. Despite the conservation in their structures and transcriptional activities, these isoforms are expressed in different tissues and have distinct roles in embryonic development and epithelial maintenance [57]. In stratified squamous epithelium, ΔNp63 is strongly expressed in basal cells where TAp63 is weak, indicating that ΔNp63 may play a major role in these epithelia. Interestingly, ΔNp63 null mice manifest with limb truncation, orofacial malformation and the defective maturation of stratified epithelia, while no obvious epidermal defects were noted in mice with targeted ablation of TAp63 [58]. Additionally, the loss of ΔNp63 induces the dysregulation of NOTCH and TGF-β signalling, linking ΔNp63 to epithelial cell fate specification and stem cell maintenance, and highlighting the notion that TP63 is the “guardian of the epithelial lineage” [59]. On the other hand, TP63 facilitates squamous cell carcinoma (SCC) formation and an in vivo deletion of the *TP63* gene in established SCC tumours leads to rapid tumour shrinkage, revealing a crucial function for TP63 in SCC maintenance [60]. Moreover, up to 80% of OSCC patients show overexpression and/or genomic amplification of *TP63*, and these events are associated with poor tumour differentiation and patient prognosis [61].

### 3.5. Epigenetic Regulators of the Commitment Switch to Epithelial Differentiation

During lineage specification, dynamic epigenetic modifications profoundly influence gene expression. The open chromatin conformation in stem cells enables active transcription of genes related to keratinocyte commitment and differentiation. This process is progressively accompanied by increased DNA methylation, the accumulation of various histone markers and chromatin remodelling, resulting in diminished chromatin accessibility and full inactivation of transcription in terminally differentiated cells [62]. These observations confirm an active role for the epigenome in fine tuning epithelial differentiation. An example of this process relates to EZH2, a component of the polycomb repressor complex 2 (PRC2), a key regulator of differentiation responsible for the trimethylation of lysine 27 on histone H3 (H3K27me3) in gene repression. EZH2 is highly expressed in basal progenitor cells and its expression correlates with cell proliferation and is thus gradually suppressed with the occurrence of epithelial differentiation. EZH2-induced H3K27me3 marker inhibits the binding of AP1 transcription activator to its target genes, leading to the repression of AP1-transcribed differentiation genes in basal cells. In addition, the loss of EZH2 in vitro and in vivo correlates with reduced proliferation of basal cells and accelerated differentiation suprabasally [63]. It was also shown that EZH2 is overexpressed in OSCC cell lines and primary tumours, and that a genetic or pharmacological inhibition of EZH2 attenuates tumour growth and restores differentiation gene expression in differentiation refractory OSCC xenografts [64]. Arguably, driver mutations in epigenetic modulators are often retained for the entire course of carcinogenesis, implying that their dysregulation is key to the loss of differentiation in OSCC tumourigenesis and an exciting area for therapeutic targeting.

Another level of regulation that follows the transcriptional control of differentiation genes depends on the post-transcriptional modifications that also influence the dynamics of protein abundance during differentiation [65]. One of the crucial post-transcriptional mechanisms is miRNA-mediated gene silencing. Small non-coding single stranded RNAs pair to target mRNAs, inhibiting translation and inducing mRNA decay. In mammalian epidermis, it was shown that the Grainy-head like 3 (GRHL3), an evolutionarily conserved transcription factor and a master regulator of terminal keratinocyte differentiation [66,67], is repressed by oncogenic miR-21 [68,69]. Importantly, an epidermal-specific loss of key miRNA processing machinery components, such as Dicer and Dgcr8, causes epidermal dehydration, hair follicle apoptosis and neonatal lethality [70]. These results underscore the importance of regulating protein expression associated with differentiation and invite further investigations of the multi-level controls of this process.

## 4. Terminal Differentiation in OSCC

Terminal differentiation programs are fully engaged in granular and cornified layers of the oral epithelium, leading to the establishment of a functional barrier that prevents against environmental insults. This outermost layer of the dorsal tongue epithelium is formed of corneocytes embedded in a lipid matrix that contains small vesicles of cholesterol, phospholipids and ceramides. In addition to protein cross-linking, the correct presence of lipid complexes is essential to maintain barrier function, preventing dehydration and infection [6]. Nevertheless, how the loss of terminal differentiation factors affects the integrity of the epithelium and the epithelial architecture are still poorly understood.

### 4.1. ABCA12

ABCA12 regulates vesicular trafficking in terminally differentiated cells and is one of the core components that preserve lipid homeostasis [6]. The loss of *Abca12* induces a failure of extracellular lipid deposition and premature differentiation, consequently leading to hyperkeratosis and impaired barrier function [71]. In humans, germline mutations in ABCA12 are linked to Harlequin Ichthyosis, a severe inherited disease causing high neonatal death, dehydrated skin and infections [72]. Interestingly, these mutations are present in 3.7% of OSCC patients with up to 27% of tumours losing ABCA12 expression [43], further proposing the loss of ABCA12 as an initiator of OSCC development.

### 4.2. FLG

Keratins and filaggrin contribute to 80–90% of the mass of the granular layer in stratified epithelia, where they are tightly cross-linked to establish a physiological epithelial barrier. Notably, filaggrin (filament aggregation protein) serves as an indispensable molecule for the aggregation process of keratin filaments. Filaggrin derives from the proteolytic maturation of profilaggrin (FLG), an S100 fused-type protein of approximately 500 kDa, histidine-rich with tandem repeats of filaggrin stored in the membrane-less protein deposits, known as keratohyalin granules (KGs) [73]. The disassembly of KGs leads to squame formation. The maturation of FLG depends on caspase-14 (CASP14), with Casp14 knockout mice showing increased *FLG* in KGs in conjunction with a striking glossy and scurfy epidermal phenotype, water loss and increased susceptibility of ultraviolet B damage [74]. Intriguingly, a decreased expression of *FLG* has been recorded in the autosomal semi-dominant disease ichthyosis vulgaris (IV) and in atopic dermatitis (AD), where the presence of loss-of-function mutations in *FLG* also predisposes to IV and AD as well as additional allergic diseases such as asthma [75]. Collectively, these observations demonstrate an essential role for filaggrin in the formation of epithelial barrier to protect against the outer environment. It is also noteworthy that *FLG* is one of the most frequently mutated genes (13%) in OSCC [43,44], and its loss disrupts OSCC differentiation and decreases therapy response [76].

### 4.3. HRNR

HRNR is also an S100 fused-type protein encoded in the epidermal differentiation complex same as FLG. The precursor of HRNR is approximately 280 kDa with a central tandem peptide repeats domain with calcium-binding sites, located in the periphery of KGs within the upper granular and cornified layers [77]. Similar to FLG, HRNR maturation proteolysis leads to releasing amino acids as natural moisturising factors for epithelial hydration and photo-protection, and to cross-linking keratins as part of the establishment of a functional barrier. *HRNR* expression is downregulated in the skin of AD patients, and a single-nucleotide polymorphism of HRNR was identified in people vulnerable to AD [78]. Importantly, 4.1% of OSCC patients show mutations in HRNR [44]. These observations suggest that defective HRNR-induced terminal differentiation may contribute to the pathophysiology of AD and OSCC (Figure 1).

## 5. Treatments for Patients with OSCC

Most OSCC patients present with advanced-stage disease, and treatment is met with high levels of recurrence and metastasis [79,80]. Moreover, OSCC patients are at high risk of developing a second primary malignancy [81]. Conventional OSCC treatment regimens include surgery, radiotherapy, chemotherapy, immunotherapy and targeted therapy [82]. The recent success of immune checkpoint blockade in cancer underlines the clinical importance of novel immunotherapy drug regimens [83], and antibodies against CTLA-4, PD-1 and PD-L1 have revolutionised OSCC care [84]. However, immunotherapy using the immune checkpoint inhibitor Pembrolizumab resulted in a relatively low (~20%) response rate, albeit in the absence of patient stratification [85,86]. The study recruited patients with advanced solid tumours over 18 years old. The archival tumour sample was obtained before the treatment and after 9 weeks of the treatment process, irrespective of PD-L1 or HPV status. Patients received 200 mg of the drug once every 3 weeks, for 24 months, and treatment response was assessed every 8 weeks by using computed tomography or magnetic resonance imaging. A significant proportion of these patients developed adaptive resistance due to the upregulation of additional immune checkpoints, whilst others experienced increased tumour growth kinetics (hyper-progressive disease) [87]. Nevertheless, OSCC with high PD-L1 expression were the most responsive, prompting the search for additional biomarkers, beyond PD-L1, that are still needed to inform the choice of therapy [88]. The interference with cancer cell-intrinsic signalling pathways was shown to modulate cancer sensitivity to immunotherapy [89]. While EGFR overexpression directly regulates immune checkpoint molecule expression and response to immunotherapy [90], the loss of *TP53* may render cancer cells resistant to T cell-mediated killing [91]. This suggests that combining immunotherapy with inhibitors of oncogenic signalling may provide greater therapeutic benefit and a rationale for a tailored OSCC-personalised targeted therapy. This approach is being investigated in a phase II clinical trial (NCT03544723) of the combination of an adenoviral p53 (Ad-p53) gene therapy administered intra-tumourally with approved immune checkpoint inhibitors in patients with recurrent or metastatic solid tumours [92], and has potential application in head and neck cancers.

## 6. Targeted Therapy against OSCC

The central concept of targeted therapy typically involves clinical testing to determine features of a patient’s malignant disease that may inform treatment decisions. Why some patients respond well to targeted therapies, and other patients who appear to have the same type of cancer respond poorly or not at all, remains poorly understood. Genomic biomarkers paired with targeted therapies have proven highly efficacious in some cases, such as HER2 amplification and trastuzumab in breast cancer [93], BRAF mutation and combined BRAF/MEK inhibition in melanoma [94] and EML4-ALK fusions and Crizotinib in lung adenocarcinoma [95]. While many studies have focused on targeting oncogenic mechanisms, none have proven totally effective for predicting responsiveness. The resistance to targeted therapy can operate at the genomic level in many cancers. Key examples include EGFR-T790M mutations and resistance to EGFR inhibitors in EGFR-mutant lung cancer [96], ESR1 mutations in estrogen receptor positive breast cancer treated with endocrine therapy [97], and reversions of pathogenic mutations in BRCA1 and BRCA2 deficient cancers treated with PARP inhibitors [98]. However, no genetic tests are routinely incorporated into the management of OSCC, and patient stratification is largely done based on clinical features, even in the absence of HPV status, with a huge knowledge gap in response biomarkers [99,100]. One of the best attempts at treating recurrent or metastatic OSCC patients with targeted therapy was made using Cetuximab, a U.S. Food and Drug Administration (FDA)-approved monoclonal antibody that specifically binds and inhibits the activity of EGFR. Despite EGFR overexpression in ~90% of OSCC patients [101], only 10% of patients derived a beneficial response to combined Cetuximab and radiotherapy while the remainder were at higher risk of relapse [102]. In recent years, it has become clear that the prognostic value of EGFR overexpression or increased gene copy number does not correlate with Cetuximab response due to common alterations downstream of EGFR [103]. Somatic mutations, genetic and epigenetic alterations have been shown to drive senescence bypass, proliferation, continuous cancer cell survival and OSCC treatment resistance [25,104]. Such mechanisms render most OSCC patients difficult to cure and emphasise the urgent need to identify additional strategies to enhance therapy response [105].

## 7. Differentiation-Paired Targeted Therapy for OSCC

Large-scale genomic and transcriptomic sequencing of OSCC tumours showed very high (~90%) inactivating mutations in tumour suppressor genes, most of which encoded terminal differentiation factors [106]. TP53 was the most dominant (>74%) mutated gene while mutations in squamous differentiation factors (e.g., TP63, Notch1, IRF6 and RIPK4) were commonly observed and co-existed in the same cancers [106,107,108]. These mutations are likely to drive more proliferative basal-like OSCC phenotypes and correlate with poor patient survival [107,108]. Surprisingly, the incidence of oncogene-activating mutations was low (~20%) and suggests that the dysregulation of differentiation may act as the main driver of TP53 mutant OSCC.

The lack of effective targeted therapies for heterogeneous OSCC, particularly those with TP53 mutations, has hampered improvements in patient survival, which has remained virtually unchanged for the last 30 years [109]. Nonetheless, recombinant human p53 adenovirus vectors (such as Ad5RSV-p53 and AdCMV-p53) have been used to replace mutated *TP53* with wild-type gene in order to restore p53 functionality, with potential utility as a new treatment approach for head and neck cancers [110,111,112]. While much has been learned about the mutational landscape of OSCC (Table 1) through next-generation sequencing [113], a tremendous challenge remains in translating this genomic information into functional outcomes [106]. Integrated approaches leveraging both genetic and epigenetic data may determine whether functional differentiation factors affect the responses to targeted therapy. Since key terminal differentiation effectors have not been explored with regards to squamous differentiation and therapy response in OSCC, particularly in those exposed to differing aetiologies, novel strategies that combine targeted therapy with terminal differentiation could lead to optimizing patients’ therapy response in a manner that is superior to traditional oncogene-targeting approaches. Innovative screening methods to stratify OSCC patients into specific subsets should enhance clinical outcomes for targeted therapies [114]. Of these, a potent switch that remains active and controls progenitor commitment to terminal differentiation and therefore induces growth arrest should be on the horizon. Such findings could open-up novel avenues for more accurate differentiation-guided treatment stratification of OSCC and could contribute to a strong evidence-based foundation for novel clinical applications.

## 8. Conclusions

OSCC genomic alterations are dominated by the loss of terminal differentiation tumour suppressor genes, with 80% of patients harbouring at least one genomic alteration in a targetable gene [104]. This suggests that novel approaches to treatment may be possible for OSCC, particularly by identifying upstream signalling leading to the induction of functional terminal differentiation factors and subsequently, OE terminal differentiation to promote therapy response.

## Figures and Tables

**Figure 1 cancers-13-05123-f001:**
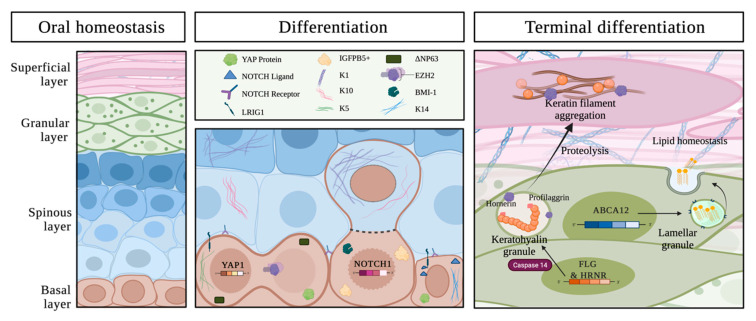
Schematic representation (original figure) of the oral layers in homeostasis (**left**) with a highlight of the basal layer depicting major players in epithelial differentiation (**middle**) and those regulating terminal differentiation in the superficial layer (**right**).

**Table 1 cancers-13-05123-t001:** Frequency of genetic alterations targeting differentiation and terminal differentiation genes in HNSCC (original table).

Name of Genes	Frequency of Genetic Alterations in HNSCC	Functions in Epithelia	Evidence	Ref
*ASXL1*	Mutation, 2.90% CNA, 2.70%	Differentiation	IMP	[115]
*DLG5*	Mutation, 2.30%	Differentiation, cell polarity	IMP	[116]
*DMBT1*	Mutation, 2.70%	Differentiation, innate immunity	IDA	[117]
*ERBB4*	Mutation, 4.50%	Differentiation	IMP	[118]
*FAT1*	Mutation, 21.60%; CNA, 6.80%	Differentiation, hippo/Wnt signalling, EMT		[10,54,55]
*GATA4*	CNA, 3.10%; *HOMDEL*	Differentiation, transcription factor	IMP	[119]
*MEF2C*	Mutation, 2.70%	Differentiation, transcription factor, histone deacetylase	IMP	[120]
*MYO9A*	Mutation, 2.90%	Differentiation, cell junction	ISO	[121]
*NOTCH1*	Mutation, 17.10%; Fusion, 0.60%	Differentiation, notch pathway		[10,36,106,107,108,122]
*NOTCH2*	Mutation, 3.90% CNA, 2.50%	Differentiation, notch pathway		[10,36,43,44]
*NOTCH4*	Mutation, 1.90%	Differentiation, notch pathway		[10,36,106,108]
*NUMA1*	Mutation, 3.70% CNA, 8.10%	Differentiation, asymmetric cell division	IMP	[122]
*ONECUT2*	CNA, 2.90% *HOMDEL*	Early differentiation, specification	IMP	[123]
*PTCH1*	Mutation, 2.70%	Differentiation, SHH signalling	IGI	[124]
*RARG*	Mutation, 2.50%	Differentiation, specification, non-cornified	IGI	[125]
*RHOA*	Mutation, 1.90%	Differentiation, specification, cell junction	ISO	[126]
*ROCK1*	Mutation, 2.90% Fusion, 0.60% CNA, 1.90%	Differentiation, polarization	IGI	[127]
*ROCK2*	Mutation, 2.10%	Differentiation, polarization	ISO	[128]
*ROS1*	Mutation, 5.00%	Differentiation	IMP	[129]
*SCRIB*	Mutation, 2.30% CNA, 2.90%	Differentiation, polarization	TAS	[130]
*SEC24B*	Mutation, 1.90%	Differentiation, polarization	IMP, IGI	[131]
*SMAD4*	Mutation, 2.90% CNA, 3.50%	Differentiation, transcription factor	IMP	[132]
*TJP1*	Mutation, 1.90%	Differentiation, tight junction protein	IBA	[133]
*TP63*	Mutation, 2.30% CNA, 16.10%	Differentiation, transcription factor		[10,57,58,59,60,61,106,108]
*TRIOBP*	Mutation, 1.90%	Differentiation, junction, AJ formation	IMP	[134]
*AGR2*	CNA, 1.90% *AMP*	Differentiation	IDA	[135]
*DLX5*	CNA, 4.10% *AMP*	Differentiation, transcription factor	IGI	[136]
*DLX6*	CNA, 4.10% *AMP*	Differentiation, transcription factor	IGI	[136]
*EHF*	CNA, 2.70% *AMP*	Differentiation, transcription factor	IEA	[137]
*ELF5*	CNA, 2.50% *AMP*	Differentiation, anti-EMT	IGI	[138]
*ESRP1*	CNA, 2.70% *AMP*	Differentiation, splicing	IMP	[139]
*EXT1*	CNA, 5.80% *AMP*	Differentiation, mesenchymal development, regeneration	IMP	[140]
*FAM20C*	CNA, 1.90% *AMP*	Differentiation, secreted phosphoproteome, wound healing	IDA	[141]
*FOXL2*	CNA, 5.00% *AMP*	Differentiation, transcription factor	IMP	[142]
*GSK3B*	CNA, 2.90% *AMP*	Differentiation, notch pathway		[69]
*IFNG*	CNA, 4.10% *AMP*	Differentiation, polarization	ISO	[143]
*KLF5*	CNA, 2.90% *AMP*	Proliferation, early differentiation	IMP	[144]
*NFIB*	CNA, 3.30% *AMP*	Mesenchymal to epithelial differentiation	IMP	[145]
*NKX2-1*	CNA, 1.90% *AMP*	Differentiation, transcription factor	IGI	[146]
*OVOL1*	CNA, 4.10% *AMP*	Differentiation, transcription factor	IBA	[133]
*PGR*	CNA, 3.90% *AMP*	Differentiation	IMP	[147]
*RFX3*	CNA, 3.70% *AMP*	Differentiation, specification, transcription factor	IMP	[148]
*SOX17*	CNA, 2.90% *AMP*	Differentiation, specification	IGI	[149]
*TBX1*	CNA, 2.10% *AMP*	Differentiation, adhesion	IMP	[150]
*ABCA12*	Mutation, 3.70%	Terminal differentiation, lipid homeostasis		[6,43,71,72]
*FLG*	Mutation, 13.00%	Terminal differentiation, cornified envelop		[43,44,73,74,76,107]
*HRNR*	Mutation, 4.10%	Terminal differentiation, cornified envelop		[43,44,77,78,107]
*MYO7A*	Mutation, 1.90% CNA, 3.10%	Terminal differentiation	IMP	[151]

The genetic alteration data were extracted from HNSCC patients (*n* = 515) available through The Cancer Genome Atlas (TCGA, PanCancer Atlas). Three housekeeping genes were used to establish the percentage of base-level mutation—ACTB: 0.8%; GAPDH: 0.4%; HPRT: 0.4%. All genes with a frequency of genetic alterations that is equal or below 0.8% were omitted. References are evidence for the gene function in epithelia, suggesting a differentiation/terminal differentiation role for the selected genes with significant genetic alterations in HNSCC. Abbreviations: AMP, amplification; CNA, copy number alteration; IDA, inferred from direct assay; IMP, inferred from mutant phenotype; ISO, inferred from sequence orthology; IGI, inferred from genetic interaction; TAS, traceable author statement; IEA, inferred from electronic annotation; IBA, inferred from biological aspect of ancestor.

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
