# Peer review of "The Balance between Differentiation and Terminal Differentiation Maintains Oral Epithelial Homeostasis"

_cancers, 2021, doi:10.3390/cancers13205123_

Round 1
Reviewer 1 Report
In this paper, the authors conducted a systematic review on the balance between differentiation and terminal differentiation in maintaining oral homeostasis and preventing epithelial transformation in oral squamous cell carcinoma (OSCC). They also discussed targeted therapy in OSCC, highlighting the current mismatch between genomic information available and ad-hoc treatments.
The topic is interesting and the review exhaustive; nonetheless, some remarks need to be made.
Major points
- The paper is rich but, at the same time, difficult to follow in all its parts by the reader. I suggest that the Authors shorten and make clearer the paragraphs on signaling pathways, and that they merge the paragraphs on targeted therapies in OSCC.
Minor points
- All the abbreviations used in the main manuscript should be written in full when they are first quoted.
- Table 1 should be integrated with the number of patients studied in each reported article.
- The bibliography should conform to the editorial guidelines of the journal (e.g. [1,4] not (1), (4)).
- The sentence starting on line 48 (OSCC is a malignant tumour…) should be rephrased: OSCC is not a disease spectrum itself but rather the final step of this evolving process.
Author Response
We are very thankful to the Reviewer for assessing our review and for providing constructive comments.
Major Issues
1. There is no mention of senescence which breaks down in ‘virtually all HPV -negative head & neck tumours by the disabling of p16INK4A and p53 ‘plus telomerase reactivation and HPV E6 and E7 disable the same pathways by degrading p53, pRB and suppressors of TERT expression (The Cancer Genome Atlas Network Nature 2015). Gene copy number variations are tightly associated with the bypass of senescence and crisis (Veeramachaneni et al Sci. Rep. 2019) and so this should be acknowledged first briefly before proceeding to discuss the role of differentiation genes. Although the recent Science paper by Loganathan et al 2020 highlighting NOTCH signalling defects as common in SCC-HN (60%) is persuasive, it is largely based on mouse models which have differences from human SCC in a number of respects (notably an absence of telomerase deregulation and regulation of the INK4A locus). In short are the mutations and CNVs in table 1 additive to senescence bypass? This needs to be briefly discussed.
We thank the Reviewer for drawing our attention to this interesting point. We have now discussed the role of senescence bypass prior to epithelial transformation briefly and cited the paper from Veeramachaneni et al., Sci. Rep. 2019. Since the frequency of genetic alterations was extracted from TCGA data, we will need to have a bioinformatic approach to ascertain whether the alterations occur after senescence bypass, before or simultaneously. To remain within the scope of the review, we have mentioned that some mutations, particularly in the NOTCH pathway, could be involved in senescence bypass (K. J. 2019, Cell Death and Differentiation).
2. P16INK4A gets a low mention yet it is mutated, deleted and methylated in just as many tumours as p53 and has nothing to do with keratinocyte differentiation as far as I am aware. P16INK4A is lost before other tumour suppressors, is not induced by calcium in simple keratinocyte cultures (Loughran et al Oncogene 1996) and has no effect on differentiation in 3D organotypic cultures (Dickson et al MCB 2000). Instead p16INK4A induction is linked to the breakdown of the basal layer, abnormal laminin gamma 5 gamma 2 expression and migration seen in carcinoma-in-situ (Natarajan et al Am. J. Pathol. 2003; 2006). This could be incorporated into the answer to point 1.
We thank the Reviewer for this great suggestion. We have discussed the role of p16INK4A activation on line 133 and added the references suggested by the Reviewer.
3. The term tumour suppressor is used somewhat loosely. Is there any evidence from human genetics studies that any of the genes discussed pre-dispose to human cancer e.g. stop codons in FAT1 or NOTCH? If so such references should be included. A true tumour suppressor is a gene that when mutated predisposes to cancer. TP53 and RB1 are tumour suppressor genes and positive regulators of telomerase are oncogenes e.g. CMYC (Wang et al Genes and Development 1999).
We have added a reference to FAT1 mutations in OSCC on line 222 and multiple references for NOTCH1 as detailed in the response to point 1.
4. It is not clear in which keratinocytes the NOTCH mutations are located and their functional consequences may be diverse. Loss of function NOTCH1-3 can lead to a tumour promoting microenvironment Dehmeri et al Cancer Cell 2009), thus explaining why keratinocytes with NOTCH1 mutations increase with age in the human oesophagus and even faster in drinkers and smokers but decline in the cancers (Martincorena et al Science 2018) where NOTCH1may well have a different tumour suppressor type function in immortal cell lines (Pickering et al Cancer Discovery 2013). Genes deregulating differentiation are rarely mutated in keratinocytes from dysplasia kerratinocytes that are still genetically stable but NOTCH1 mutations are sometimes detected (Veeramachaneni et al Sci. Rep. 2019; DeBoer et al Mol. Cancer Res. 2019). In a total of 24 such cultures only two NOTCH1 mutations were detected and no mutations of AJUBA, TP63, FAT1, NOTCH1, NOTCH2 or NOTCH3. This needs to be mentioned and briefly discussed.
We appreciate the Reviewer’s comment. We have included additional information and discussed the role of NOTCH1 in more details on lines 183, 192, 198 and 206 with the relevant references.
5. Table 1 Needs some editing/refining and the evidence for CNVs and homozygous deletions need to be checked for the following.
Is there any functional evidence to back up the genetics? Are the gene gains and deletions focal? For example are the deletions intragenic? We thank the Reviewer for this important comment. The mutant genes were selected from TCGA with a significant frequency of intragenic alterations (higher than three standard housekeeping genes). These genes are confirmed to play roles in differentiation/terminal differentiation with corresponding references supporting their roles. In keeping with the main message of this review, we propose a link between genes with significant genetic alterations in HNSCC and their established role suggesting an impairment of differentiation/terminal differentiation in HNSCC.
Are the gene gains in amplicons or high copy number gains? The genetic alteration data were extracted from HNSCC patients (n=515) available through The Cancer Genome Atlas (TCGA, PanCancer Atlas). The frequency of alterations is the percentage of events in the total number patients. As the Reviewer would expect, the copy number gains, loss or invariant are variable across the selected genes in different patients due to multiple reasons including genetic interactions and non-genetic factors. Nevertheless, the take-home message of the table is the link between mutant genes and epithelial differentiation/terminal differentiation including the head and neck.
How do these frequencies compare to changes in housekeeping genes? We thank the Reviewer for this important point. We have now identified the frequency of mutations for three housekeeping genes (ACTB, HPRT, GAPDH) and omitted from the table all genes with a frequency of genetic alterations that is equal or below 0.8%, the highest base level mutation of a housekeeping gene (ACTB). This information is now added in the table legend.
This needs to be clarified and the authors need to be careful not to over-interpret or misrepresent evidence. TP63 copy number gains are only in immortal cultures and part of a large region of gain (Veeramachaneni et al 2019)
Detailed Comments
1. I am not up-to-date with the latest stem cell concepts but am puzzled by section 2.1. Did not lineage tracing experiments lead the Clevers and Jones groups to conclude that the ultimate permanent residents of the intestinal and skin epithelia were fast cycling (Barker et al Nature 2007) and that there was ‘no such thing as a professional hard-wired stem cell’ with only a few exceptions (Clevers Science 2015) and ‘also transit amplifying cells (Jones and Watt Cell Stem Cell 2007). Please expand this section and update it.
This information has been added on line 95 with a reference to Clevers H, Science 2015.
2. Section 2.3 Cancer stem cells are a moving target (see also above) and recent evidence suggests that they are increased following the induction of senescence by cancer therapy (Milanovich et al Nature 2018). Where do the authors think CSCs originate?
We have added additional details and references on cancer stem cells as suggested by the Reviewer on line 142. Cancer stem cells (CSCs) are believed to act as the drivers of OSCC. These cells are highly tumorigenic with increased self-renewal and clonal capacities. They may originate from the accumulation of multiple oncogenic hits in normal stem cells, in differentiated or senescent cells that result in their reprogramming into CSC.
3. Section 3.2 Notch was originally not considered to be a great drug target because inhibiting NOTCH causes skin cancer in clinical trials (Extance et al Nature Drug Discovery 2010) and this may be because loss of function NOTCH can lead to a tumour promoting microenvironment (see above). More selective gamma secretase inhibitors have recently been reported (Habets et al Sci.Trans.Med.2019) but maybe more detail surrounding the role of NOTCH signalling in oral SCC would make this clearer. NOTCH is complicated and maybe some more insight into activating versus loss of function mutations would be helpful and how the recent 2017 paper (ref 34) relates to this. Can the authors clarify this area and discuss?
We thank the Reviewer for this comment. The information is now clarified further on line 183, 192, 198 and 206, with additional references cited.
4. Section 3.3 and 3.4 and elsewhere: gene CNVs are post senescence breakdown and gene amplification is regulated by p53 which is mutated in at least 97% of SCC-HN tumours (The Cancer Genome Atlas Network Nature 2015) and nearly all immortal SCC lines . Also, over-expression data is weak evidence of causality and this needs to be born in mind.
Same as point 1, the paper from Veeramachaneni et al., Sci. Rep. 2019 has been cited.
5. The authors are largely right in their statements in sections 7 and 8 that targeting oncogenic signalling pathways has been very disappointing and it is not clear why. However, recent organoid data from the Clevers group (Driehuis et al Cancer Discovery 2019) claims that patients who do not respond well to cetuximab have downstream mutations in a subset of tumour cells. However, the genetics of the SCC-HN organoids in the Driehuis study are not the same as SCC-HN tumours in vivo, particularly as regards EGFR status. Perhaps this study should be cited.
The study by Driehuis et al., has been added as a reference on line 382.
Lines 371 to 376 The restoration of p53 in animal models causes tumour regression by (inducing senescence (Xu et al Nature 2007) provided the immune system is intact and they suggested induction of senescence and or differentiation therapies. This original suggestion should be acknowledged.
The reference from Xu et al., Oncogene 2008 has been added.

Reviewer 2 Report
Strengths
In general, the review by Bai et al is very well written especially the portion on normal oral keratinocyte differentiation. As regards cancer the review is focused on the role of genes that regulate differentiation in oral epithelial homeostasis and cancer. It will be of interest to the readership of ‘Cancers’.
Weaknesses
The article does not put the cancer sections into any real context with regard to the stage at which differentiation breakdown takes place and how these events relate to the breakdown of senescence and genetic instability which are ubiquitous in squamous cell carcinomas of the head and neck (see below). The data in Table 1 may be over-stated and some caution is required in interpreting gene copy number variations in the absence of intervention e.g. the study by Loganathan et al Science 2020 which highlighted the haploinsuffiency of several low impact NOTCH pathway mutations.
Major Issues
- There is no mention of senescence which breaks down in ‘virtually all HPV -negative head & neck tumours by the disabling of p16INK4A and p53 ‘plus telomerase reactivation and HPV E6 and E7 disable the same pathways by degrading p53, pRB and suppressors of TERT expression (The Cancer Genome Atlas Network Nature 2015). Gene copy number variations are tightly associated with the bypass of senescence and crisis (Veeramachaneni et al Sci. Rep. 2019) and so this should be acknowledged first briefly before proceeding to discuss the role of differentiation genes. Although the recent Science paper by Loganathan et al 2020 highlighting NOTCH signalling defects as common in SCC-HN (60%) is persuasive, it is largely based on mouse models which have differences from human SCC in a number of respects (notably an absence of telomerase deregulation and regulation of the INK4A locus). In short are the mutations and CNVs in table 1 additive to senescence bypass? This needs to be briefly discussed.
- P16INK4A gets a low mention yet it is mutated, deleted and methylated in just as many tumours as p53 and has nothing to do with keratinocyte differentiation as far as I am aware. P16INK4A is lost before other tumour suppressors, is not induced by calcium in simple keratinocyte cultures (Loughran et al Oncogene 1996) and has no effect on differentiation in 3D organotypic cultures (Dickson et al MCB 2000). Instead p16INK4A induction is linked to the breakdown of the basal layer, abnormal laminin gamma 5 gamma 2 expression and migration seen in carcinoma-in-situ (Natarajan et al Am. J. Pathol. 2003; 2006). This could be incorporated into the answer to point 1.
- The term tumour suppressor is used somewhat loosely. Is there any evidence from human genetics studies that any of the genes discussed pre-dispose to human cancer e.g. stop codons in FAT1 or NOTCH? If so such references should be included. A true tumour suppressor is a gene that when mutated predisposes to cancer. TP53 and RB1 are tumour suppressor genes and positive regulators of telomerase are oncogenes e.g. CMYC (Wang et al Genes and Development 1999).
- It is not clear in which keratinocytes the NOTCH mutations are located and their functional consequences may be diverse. Loss of function NOTCH1-3 can lead to a tumour promoting microenvironment Dehmeri et al Cancer Cell 2009), thus explaining why keratinocytes with NOTCH1 mutations increase with age in the human oesophagus and even faster in drinkers and smokers but decline in the cancers (Martincorena et al Science 2018) where NOTCH1 may well have a different tumour suppressor type function in immortal cell lines (Pickering et al Cancer Discovery 2013). Genes deregulating differentiation are rarely mutated in keratinocytes from dysplasia kerratinocytes that are still genetically stable but NOTCH1 mutations are sometimes detected (Veeramachaneni et al Sci. Rep. 2019; DeBoer et al Mol. Cancer Res. 2019). In a total of 24 such cultures only two NOTCH1 mutations were detected and no mutations of AJUBA, TP63, FAT1, NOTCH1, NOTCH2 or NOTCH3. This needs to be mentioned and briefly discussed.
- Table 1 Needs some editing/refining and the evidence for CNVs and homozygous deletions need to be checked for the following.
Is there any functional evidence to back up the genetics? Are the gene gains and deletions focal? For example are the deletions intragenic?
Are the gene gains in amplicons or high copy number gains?
How do these frequencies compare to changes in housekeeping genes?
This needs to be clarified and the authors need to be careful not to over-interpret or misrepresent evidence. TP63 copy number gains are only in immortal cultures and part of a large region of gain (Veeramachaneni et al 2019)
Detailed Comments
- I am not up-to-date with the latest stem cell concepts but am puzzled by section 2.1. Did not lineage tracing experiments lead the Clevers and Jones groups to conclude that the ultimate permanent residents of the intestinal and skin epithelia were fast cycling (Barker et al Nature 2007) and that there was ‘no such thing as a professional hard-wired stem cell’ with only a few exceptions (Clevers Science 2015) and ‘also transit amplifying cells (Jones and Watt Cell Stem Cell 2007). Please expand this section and update it.
- Section 2.3 Cancer stem cells are a moving target (see also above) and recent evidence suggests that they are increased following the induction of senescence by cancer therapy (Milanovich et al Nature 2018). Where do the authors think CSCs originate?
- Section 3.2 Notch was originally not considered to be a great drug target because inhibiting NOTCH causes skin cancer in clinical trials (Extance et al Nature Drug Discovery 2010) and this may be because loss of function NOTCH can lead to a tumour promoting microenvironment (see above). More selective gamma secretase inhibitors have recently been reported (Habets et al Sci.Trans.Med.2019) but maybe more detail surrounding the role of NOTCH signalling in oral SCC would make this clearer. NOTCH is complicated and maybe some more insight into activating versus loss of function mutations would be helpful and how the recent 2017 paper (ref 34) relates to this. Can the authors clarify this area and discuss?
- Section 3.3 and 3.4 and elsewhere: gene CNVs are post senescence breakdown and gene amplification is regulated by p53 which is mutated in at least 97% of SCC-HN tumours (The Cancer Genome Atlas Network Nature 2015) and nearly all immortal SCC lines . Also, over-expression data is weak evidence of causality and this needs to be born in mind.
- The authors are largely right in their statements in sections 7 and 8 that targeting oncogenic signalling pathways has been very disappointing and it is not clear why. However, recent organoid data from the Clevers group (Driehuis et al Cancer Discovery 2019) claims that patients who do not respond well to cetuximab have downstream mutations in a subset of tumour cells. However, the genetics of the SCC-HN organoids in the Driehuis study are not the same as SCC-HN tumours in vivo, particularly as regards EGFR status. Perhaps this study should be cited.
Lines 371 to 376 The restoration of p53 in animal models causes tumour regression by (inducing senescence (Xu et al Nature 2007) provided the immune system is intact and they suggested induction of senescence and or differentiation therapies. This original suggestion should be acknowledged.
Author Response
We would like to thank Reviewer #2 for assessing our paper and for their positive feedback.
This review has discussed the molecular differences between oral cell proliferation, differentiation and terminal differentiation. It has also considered the importance of these phenomena and associated molecular genetics in the context of clinical management of oral cancer. Overall, this review summarises the current state of the literature well and highlights numerous gaps in knowledge. It is very well written and will be of great interest to the head and neck cancer community.
Minor comments:
Section 2, line 89: please remove ‘the’ from the following sentence. Over the last few decades, several techniques have been developed to identify and characterise the oral stem cells under different physiological and pathological conditions. This has been removed.
Section 2.1, line 100: please use italics for Bmi1. This is corrected.
Section 3.4, line 209: Please replace the first use of ‘have’ in the following sentence. ‘Despite the conservation in their structures and transcriptional activities, these isoforms are expressed in different tissues and have shown to have distinct roles in embryonic development and epithelial maintenance (42). This is corrected.
Section 5. Please ensure that this section is clear with regards to the setting in which Pembrolizumab has efficacy. The clinical trials cited refer to patients with recurrent/metastatic disease. Please add the details of this group to this section for clarity. Details on the clinical trials cited have been added in section 5.
Section 6, line 341: Please add ‘have’ to the sentence ‘……. ‘none ____ proven totally effective for predicting responsiveness’. This has been added.

Reviewer 3 Report
This review has discussed the molecular differences between oral cell proliferation, differentiation and terminal differentiation. It has also considered the importance of these phenomena and associated molecular genetics in the context of clinical management of oral cancer. Overall, this review summarises the current state of the literature well and highlights numerous gaps in knowledge. It is very well written and will be of great intereCCOSst to the head and neck cancer community.
Minor comments:
Section 2, line 89: please remove ‘the’ from the following sentence. Over the last few decades, several techniques have been developed to identify and characterise the oral stem cells under different physiological and pathological conditions.
Section 2.1, line 100: please use italics for Bmi1.
Section 3.4, line 209: Please replace the first use of ‘have’ in the following sentence. ‘Despite the conservation in their structures and transcriptional activities, these isoforms are expressed in different tissues and have shown to have distinct roles in embryonic development and epithelial maintenance (42).
Section 5. Please ensure that this section is clear with regards to the setting in which Pembrolizumab has efficacy. The clinical trials cited refer to patients with recurrent/metastatic disease. Please add the details of this group to this section for clarity.
Section 6, line 341: Please add ‘have’ to the sentence ‘……. ‘none ____ proven totally effective for predicting responsiveness’.
Author Response
We greatly appreciate the revision by Reviewer #3 and their positive feedback.
In this paper, the authors conducted a systematic review on the balance between differentiation and terminal differentiation in maintaining oral homeostasis and preventing epithelial transformation in oral squamous cell carcinoma (OSCC). They also discussed targeted therapy in OSCC, highlighting the current mismatch between genomic information available and ad-hoc treatments.
The topic is interesting and the review exhaustive; nonetheless, some remarks need to be made.
Major points
- The paper is rich but, at the same time, difficult to follow in all its parts by the reader. I suggest that the Authors shorten and make clearer the paragraphs on signaling pathways, and that they merge the paragraphs on targeted therapies in OSCC.
We appreciate the suggestion from the Reviewer however, we are not able to shorten the paragraph on signalling pathways without compromising our answers to some of the questions raised by Reviewer #1, which are embedded in section 3.
Minor points
- All the abbreviations used in the main manuscript should be written in full when they are first quoted. The review has been re-checked and corrected for the abbreviations.
- Table 1 should be integrated with the number of patients studied in each reported article. The genetic alteration data were extracted from HNSCC patients (n=515) available through The Cancer Genome Atlas (TCGA, PanCancer Atlas). This information has been added to the table caption.
- The bibliography should conform to the editorial guidelines of the journal (e.g. [1,4] not (1), (4)). This has been corrected.
- The sentence starting on line 48 (OSCC is a malignant tumour…) should be rephrased: OSCC is not a disease spectrum itself but rather the final step of this evolving process. This has been corrected.

Round 2
Reviewer 1 Report
The authors have modified and revised the article taking on board most of our comments; I think the article is now much more fluent and its various components are more connected and consequential. For me it is publishable.Author Response
We thank the Reviewer for accepting our revision and approving the publication of our review.

Reviewer 2 Report
The authors have largely answered my major concerns, especially those regarding the data in Table 1.
I have a few minor points:
Original comment:
The term tumour suppressor is used somewhat loosely. Is there any evidence from human genetics studies that any of the genes discussed pre-dispose to human cancer e.g. stop codons in FAT1 or NOTCH? If so such references should be included. A true tumour suppressor is a gene that when mutated predisposes to cancer. TP53 and RB1 are tumour suppressor genes and positive regulators of telomerase are oncogenes e.g. CMYC (Wang et al Genes and Development 1999).
Answer:We have added a reference to FAT1 mutations in OSCC on line 222 and multiple references for NOTCH1 as detailed in the response to point 1.
There is a misunderstanding here. Somatic mutations are not evidence that the genes suppress cancer. To be strict unless there is evidence that NOTCH and FAT loss of function in the human germline pre-disposes to cancer like APC, TP53. CDKN2A, BRCA1/2, ATM etc. the use of the term driver mutations would be better. Please re-word.
The reference from Xu et al., Oncogene 2008 has been added.
This is not the right reference. The correct reference is Senescence and tumour clearance is triggered by p53 restoration in murine liver carcinomas.Xue W, Zender L, Miething C, Dickins RA, Hernando E, Krizhanovsky V, Cordon-Cardo C, Lowe SW.Nature. 2007 Feb 8;445(7128):656-60. doi: 10.1038/nature05529. Epub 2007 Jan 24.
Please change.
Minor typos
I have not proof read the manuscript but on line 206/207 cautions should be caution
Please change
Author Response
We would like to thank the Reviewer for the additional minor comments.
We have now changed the mention of tumour suppressors to driver mutations as suggested. We have also replaced the wrong reference (Xu et al., Oncogene 2008) with the correct one (Xue et al., Nature 2007).
Cautions is corrected to caution.
Again, we thank the Reviewer for their time and efforts which have clearly enhanced the quality of our review.
Best wishes
